# Modified Graphene/Muscovite Nanocomposite as a Lubricant Additive: Tribological Performance and Mechanism

**Zhinan Zhao** [†]**, Yujunwen Li** [†]**, Wu Lei** *[ID] **and Qingli Hao** *

Key Laboratory for Soft Chemistry and Functional Materials, Nanjing University of Science and Technology, Ministry of Education, Xiao Ling Wei 200, Nanjing 210094, China
* Correspondence: leiwuhao@njust.edu.cn (W.L.); qinglihao@njust.edu.cn (Q.H.)
† These authors contributed equally to this work.

**Abstract:** Modified graphene/muscovite (MGMu) nanocomposite was synthesized with muscovite (Mu) and silane coupling agent modified graphene oxide through a simple hydrothermal method that exhibited excellent dispersion stability in oil. Compared with the base oil sample, the average friction coefficient and wear scar diameter of the MGMu oil sample decreased by 64.4 and 20.0%, respectively, and the microhardness of its wear scar was increased by 16.1%. The MGMu showed better tribological performance than its individual component due to the synergetic effect between the two components. The lubrication mechanism was proposed according to the morphology, chemical composition, and microhardness of the surface of wear scars. MGMu as an oil additive could fill between the friction pairs, cling to some asperities, and occur relative sliding between unit layers, thus playing a role in lubrication. It was found that MGMu would react with the surface of the friction pair during the friction process to generate $Fe_2O_3$, $SiO_2$, SiC, and new aluminosilicate, which formed a self-repairing layer with high hardness. This chemically reactive film exhibited a lower shear strength, which made the oil sample containing MGMu have a lower coefficient of friction.

**Keywords:** graphene-based composites; muscovite; tribological properties; lubricant; self-repairing layer





## 1. Introduction

Wear and tear are the main reasons for the failure of mechanical parts, and the rapid repair and remanufacturing repair of the friction and wear interface under the multi-constraint conditions of the equipment site are the major problems that technicians look forward to solving. Against this background, materials with anti-friction, anti-wear, and self-healing abilities show broad application prospects. At the same time, lubricants are widely used in various fields and are composed of base oil and additives. The properties of the additives have an important impact on the lubricating performance of the lubricating oil, and a single additive is difficult to meet a wide variety of requirements. Therefore, the research of compound lubricating oil additives becomes crucial. At the same time, the dispersibility of the oil additives is also one indicator for evaluating lubricating oils. And the additives with good dispersion stability in the lubricants can provide better lubricating properties [1]. Layered silicate minerals are low-cost two-dimensional materials widely used because of their weak bonding force in the unit layer and easy cleavage along the layer direction, which can be used as lubricating materials. In recent years, the research of silicate materials as lubricant additives has also developed rapidly [1–5].

Muscovite (Mu), whose chemical formula is $KAl_2 [AlSi_3O_{10}] (OH)_2$, belongs to a kind of layered silicate mineral. As shown in Supplementary Materials Figure S1, its unit layer structure is composed of two silicon-oxygen tetrahedral layers and an aluminum oxygen octahedral layer sandwiched in the middle. According to the characteristics of its unit layer structure configuration, it belongs to the tetrahedron-octahedron-tetrahedron (TOT) type layered silicate mineral [6]. Part of $Si^{4+}$ in the silica tetrahedron of Mu is replaced by $Al^{3+}$,

which causes the imbalance of structural charge. Therefore, there will be $K^+$ filled in to balance the charge between the layers of the unit layer. In the unit layer, the combination of the silicon-oxygen tetrahedron and aluminum oxygen octahedron is relatively firm. Still, the combination between unit layers is realized by an ion bond, which is a relatively weak force, resulting in the cleavage between layers along the direction of the $K^+$ plane. That is, relative sliding is easy to occur between unit layers, which gives Mu a certain lubrication performance [7].

Some scholars had also explored Mu as a lubricating material. Yuan et al. [8] prepared nanoparticles of muscovite. They used it as a lubricating oil additive and evaluated its anti-wear and friction reduction performances under different experimental conditions at a four-ball tester. The results showed that muscovite could improve the performance of anti-wear and friction reduction of lubricating oil. Still, the effect of the particle size, addition, and crystal type on the dispersion property should be considered. Moreover, the layered intercalation and surface modification can improve the dispersion property of muscovite nanoparticles, but it has less influence on the tribological property.

Combining with some nanomaterials, the tribological properties of Mu may be improved. For example, Du et al. [7] prepared $CeO_2$ nanoparticles and muscovite/$CeO_2$ (MC–Ce) composite through mechanical solid-state-chemistry-reaction and surface modification of oleic acid. It indicated that MC–Ce and its individual components could improve the friction-reduction and anti-wear properties of lubricant grease, and the MC–Ce composite presented the best tribological performance. The excellent tribological performance of MC–Ce may be attributed to the formation of a chemical reaction film mainly consisting of $Fe_2O_3$ and $SiO_2$, as well as the formation of an adsorption film of MC–Ce on worn surfaces. However, in practical production and application, due to the nature of hydrophilicity and oil repellent of –OH on the surface of Mu, it is difficult to uniformly and stably disperse Mu in oil. After standing for a short time, the particles tend to agglomerate and settle, which may aggravate the wear of the workpieces.

Du et al. [9] also prepared Cu-doped muscovite composite particles (Mu/Cu) via the liquid phase reduction method, and the cubic Cu nanoparticles evenly coated muscovite in composite particles. Through evaluating the tribological properties of Mu/Cu and Mu as lubricant additives in lithium grease on a block-ring tribomachine and exploring its tribological mechanism, they found that both Mu/Cu and Mu can effectively improve the tribological properties of lithium grease, and Mu/Cu exhibits better tribological performance than Mu. The friction coefficient of Mu/Cu is decreased by 69.2% compared to that of lithium grease. The layer structure of muscovite is synergistic with Cu nanoparticles in contributing to the formation of lubricant film mainly consisting of O, Si, Fe, Cu, and Al elements on the block worn surface, thereby further reducing the friction and wear.

Graphene and its derivative graphene oxide (GO) have significant lubricating potential [10–14]. However, GO does not have lipophilicity either, due to many surface oxygen-containing functional groups. In order to make it stably disperse in the oil as a lubricating oil additive, the oxygen-containing functional groups on the surface need to be modified. Herein, GO was modified and composited with Mu to prepare a series of binary composited materials for lubricating oil additives, and a four-ball friction tester investigated their tribological properties. The results show that the prepared composites have excellent lubricating properties and react with the surface elements of the friction pair to form a self-repairing layer with higher hardness.

## 2. Materials and Methods

### 2.1. Materials

Muscovite was purchased from Lingshou County Chuanshi Mineral Products Processing Factory (Shijiazhuang, China). The GO was synthesized according to a modified Hummers method in our laboratory as previous report [1,15]. Silane coupling agent (KH550, chemically pure) was purchased from Shanghai Yuanye Bio-Technology Co., Ltd. (Shanghai, China). Ethanol (analytically pure) was purchased from Sinopharm Chemical

Reagent Co., Ltd. (Shanghai, China). 15w40 oil used as base oil was purchased from China Petroleum & Chemical Corporation Lubricant Branch (Xiamen, China). Petroleum ether (60~90 °C, analytically pure) was purchased from Sinopharm Chemical Reagent Co., Ltd. (Shanghai, China). All chemical reagents were used without further purification.

### 2.2. Synthesis of the Modified Graphene/Muscovite

The synthesis process of MGMu is shown in Figure 1. Firstly, 5.33 g GO with 3 wt.% was added to 350 mL water in ultrasonication by stirring for 2 h and stirred for 3 h at room temperature to form GO suspension. Then, 15 mL silane coupling agent (KH550) was added drop by drop to the above-mentioned GO suspension. After being stirred for 1 h at room temperature, the suspension was continuously stirred for 2 h with a water bath at 80 °C. Finally, a wet sample of the modified graphene oxide (MGO) was prepared by centrifugal washing with water.

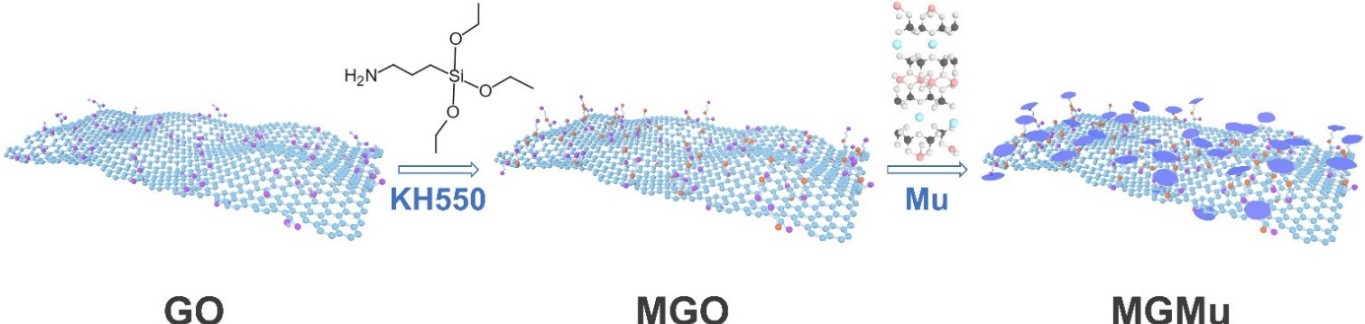

**Figure 1.** Synthesis process diagram of MGMu.

After that, 0.64 g Mu powder was weighed, poured into a ball milling jar with 10 mL of ethanol, milled for 12 h, and taken out. Next, the above-wet sample of MGO was transferred entirely into the beaker containing 350 mL water and stirred for 1 h to form the MGO suspension. Then, the entirely milled Mu was added slowly to the MGO suspension and stirred for 1 h at room temperature. Then, the above mixture was transferred into an autoclave and kept in an oven at 150 °C for 5 h under hydrothermal conditions. The product was centrifugally washed with water and collected by freeze-drying as MGO and Mu composite powder (MGMu). Finally, the MGMu powder was added to a ball milling jar with a certain amount of base oil and milled for 15 min to get a series of MGMu oil samples.

### 2.3. Characterization of MGMu

To determine the chemical composition of the materials, the analysis is carried out with the aid of Fourier transform infrared spectroscopy (FTIR Spectrometer, Nicolet IS10, Thermo Fisher, Shanghai, China) with Ever-Glo optical source (Thermo Fisher, Shanghai, China). In this work, the test samples are all in powder form, and the test range is 500–4000 cm$^{-1}$.

X-ray diffraction pattern (XRD, D8 Advance, Bruker, Billerica, MA, USA) analyses were used to characterize the material's crystal structure. The test samples are all powder, and the anode target is the Cu target Kα line (λ = 1.5406 Å), and the scanning angle is ranged from 5 to 80° of 2θ.

The material's morphology was observed using Transmission electron microscopy (TEM, TECNAI 20, FEI Company, Hillsboro, OR, USA). The powder sample was dispersed in ethanol with ultrasonication, dropped on the copper mesh, and then put into the equipment for testing after drying.

### 2.3.1. Tribological Tests

The tribological properties were mainly measured on an MRS-10G four-ball tribometer (Jinan Yongce Industrial Equipment Co., Ltd., Jinan, China) under 197 N with a rotary velocity of 600 r/min at room temperature for 1 h. The test parameters mainly refer to the

petrochemical industry standard of the People's Republic of China, NB/SH/T 0189-2017 standard test method for wear preventive characteristics of lubricating fluid—four-ball. The friction pair consists of four identical bearing GCr15 steel balls with a diameter of 12.7 mm (Shanghai Steel Ball Plant CO., Ltd., Shanghai, China), one rotating on the top while the others are fixed beneath. All test-section components were treated with ultrasonic in petroleum ether for 3 min and dried before tests. And the friction coefficient was recorded automatically using a strain sensor. Figure S2 shows the friction pair model of the four-ball friction tester.

It can be seen from the figure that the upper ball and the lower ball are in contact with each other. The contact point of the two balls undergoes local elastic deformation under the action of the pressure $W$ perpendicular to the tangent point, forming a circular contact area with a radius $R$. Since the material of the upper and lower balls is the same, that is, the Poisson's ratio ($\mu$) and the elastic modulus ($E$) of the upper and lower balls are the same, the steel ball parameters $\mu = 0.3$, $E = 2.085 \times 10^5$ Mpa can be substituted so that Hertz's formula (Equation (1)) can be simplified as follows:

$$R = 1.11\left[\frac{W}{E}\left(\frac{R_1 R_2}{R_1 + R_2}\right)\right]^{\frac{1}{3}} \tag{1}$$

$R_1$ and $R_2$ mean the radius of the upper and lower balls, respectively. Since the steel balls are the same, $R_1$ and $R_2$ are equal to 12.7 mm. At the same time, according to the model, the relationship between the normal load $W$ of the upper ball to each lower ball perpendicular to the tangent point and the load $F$ acting on the upper ball can be calculated as follows:

$$W = \frac{\sqrt{6}}{3}F \tag{2}$$

Equation (2) can be simplified, and the corresponding effect can be calculated. When the load on the upper ball is $F = 197$ N, Hertzian contact stress between the steel balls is about 2287 Mpa.

### 2.3.2. Analysis of Wear Scar Surfaces

The morphology and composition of the wear scar surfaces were analyzed by the field-emission scanning electron microscopy (SEM, JSM-IT500HR (JEOL Ltd., Tokyo, Japan)) with an energy-dispersive spectrum (EDS). After the tribological tests, the lower steel balls of the friction pair were cut and sampled, with a thickness of about 1~2 mm. The samples were washed with petroleum ether and then adhered to the sample stage with conductive adhesive for testing.

A micro Vickers hardness tester (MHVD-1000IS, Shanghai Jvjing Precision Instrument Manufacturing Co., Ltd., Shanghai, China) characterized the hardness of the steel balls' wear scars. In the test, the surface of the steel ball with the wear scar was placed on the top and fixed in the sample stage. The hardness of the wear scar surface can be obtained by converting the indentation depth of the indenter on the wear scar surface.

The wear scar diameter was measured by a 15 J optical microscopy (Shanghai Optical Instruments Sixth Factory Co., Ltd., Shanghai, China). Each wear scar was measured at least five times to guarantee the standard deviations below 5%.

X-ray photoelectron spectroscopy (XPS, PHI QUANTERA II (ULVAC JAPAN Ltd., Chigasaki, Kanagawa, Japan)) measurements were used to determine the composition and the content of the elements on the wear scars at the monochromatic Al Kα (150 W, 500 μm and 1486.6 eV) radiation.

## 3. Result and Discussion

### 3.1. Characterization of Materials

The FTIR spectra of different samples (GO, MGO, Mu and MGMu) are shown in Figure 2a. It can be seen that GO appears to have a very large broad peak at 3500–3100 cm$^{-1}$, which is attributed to the stretching vibration of –OH. It indicates that there is a certain amount of hydroxyl groups on the surface of GO, which can contribute to the hydrophilicity of GO. Therefore, GO cannot be well dispersed in oil; the subsequent results of sedimentation tests will also confirm this. The vibrational peaks at 1714, 1581, 1395, and 1146 cm$^{-1}$ in the FTIR spectra of GO correspond respectively to the carbonyl group's stretching vibration of –CO–, the asymmetric stretching vibration of C–O in –COOH, the in-plane bending vibration of CO–H and the absorption of C–O–C [16,17].

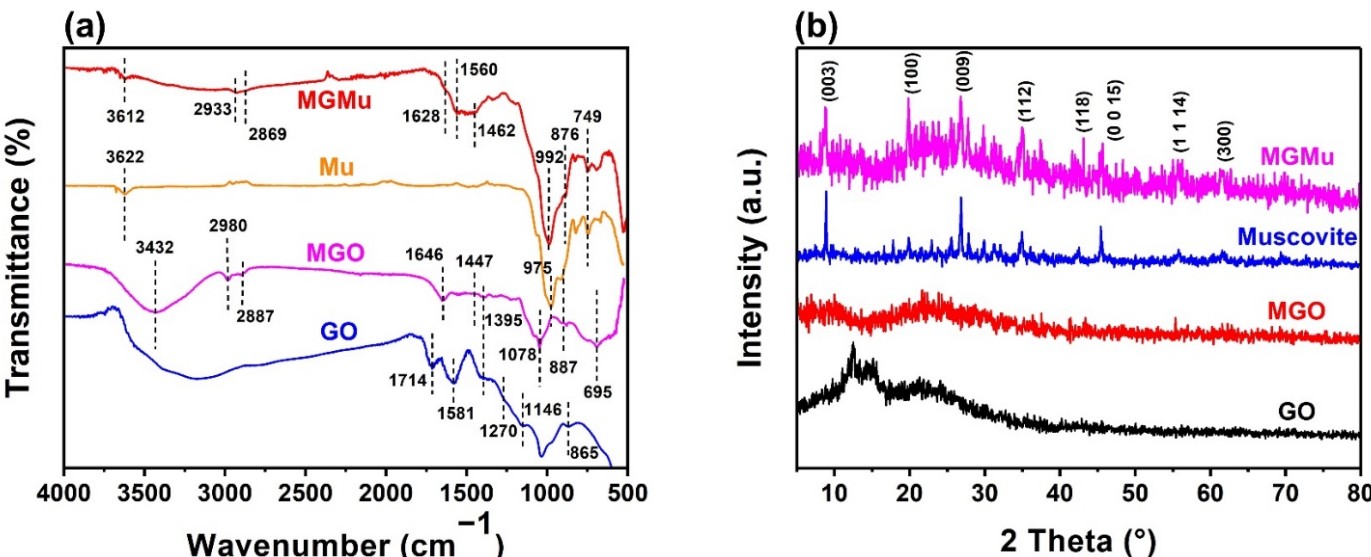

**Figure 2.** (**a**) FTIR spectra and (**b**) XRD patterns of GO, MGO, Mu and MGMu.

The peak of MGO, which was modified by adding a silane coupling agent, is located at 3432, 2980, 2887, 1646, 1078 and 887 cm$^{-1}$, respectively. To begin with, compared with GO, the broad peak located at 3500~3100 cm$^{-1}$ is much weakened. It shows that after modification, the hydroxyl groups on the surface of GO reacted with the silane coupling agent and were basically reduced. At the same time, the peaks located at 1714, 1581 and 1395 cm$^{-1}$ in GO basically disappeared, indicating that the oxygen-containing functional groups located on the surface of GO were reduced.

Next, the peaks located at 2980 and 2887 cm$^{-1}$ are the asymmetric stretching vibration peaks of the alkyl group, which is attributed to the alkyl group in the silane coupling agent, indicating that the component of KH550 does exist in MGO. Meanwhile, the peak at 887 cm$^{-1}$ represents the stretching vibration of Si–C in silane, which can also illustrate this.

Then, the peaks at 1646 and 1078 cm$^{-1}$ in MGO correspond to the in-plane bending vibration of CO–N in the amide bond and the stretching vibration of Si–O–C, respectively. It shows that after hydrolysis, the silyl group of the silane coupling agent reacted with the oxygen-containing functional groups on the surface of GO. At the same time, some amino groups of the coupling agent reacted with the carboxyl groups on the surface of GO to generate amide bonds [18,19].

Thus, the above analysis proved that KH550 reacted with GO, and the modified graphene oxide (MGO) was successfully prepared. Due to the grafting of KH550 on the surface of GO, there are long-chain alkyl groups on the surface, which greatly reduces the hydrophilicity of GO, thereby expected to improve its dispersibility in oil in further application.

As shown in Figure 2a, the FTIR spectrum of Mu is relatively simple. The peak located at 3622 cm$^{-1}$ is the stretching vibration peak of free hydroxyl groups on the surface of Mu, and the peak at 975 cm$^{-1}$ is the stretching vibration peak of Si–OH and Si–O bond of Si–O–Si in a silicon-oxygen tetrahedron.

However, in the FTIR spectrum of MGMu, compared with Mu, the peak at 3622 cm$^{-1}$ is weakened, indicating that the KH550 grafted on the surface of MGO reacted with the hydroxyl groups on the surface of Mu, and the characteristic peaks of MGO and Mu still exist in the FTIR spectrum of MGMu such as the asymmetric stretching vibration peaks of the alkyl group at 2933 and 2869 cm$^{-1}$, the bending vibration peak of the methylene group at 1462 cm$^{-1}$, and the stretching vibration peak of Si–O bond at 992 cm$^{-1}$. However, compared with the FTIR spectra of Mu and MGO, it can be seen that the above characteristic peaks are shifted to a certain extent, the alkyl peaks are red-shifted, and the peak positions of silicon-oxygen bonds are blue-shifted, which also proves that the interaction between both of them. At the same time, the peak at 876 cm$^{-1}$ indicates the existence of the Si–O bond in Si–O–C, and compared with MGO, the peak of the Si–O bond not only has a red-shift but also has a significant increase in intensity, which also proves the reaction between MGO and the hydroxyl group of Mu [20]. In conclusion, the MGMu was successfully compounded [21].

The XRD patterns of different samples are shown in Figure 2b. The XRD pattern of GO shows a strong peak at $2\theta = 12.36°$, representing the (001) crystal plane of GO, and after modification by the coupling agent. The peak of MGO located there disappears. It means that the (001) crystal plane of GO was destroyed due to the combination of the coupling agent with oxygen-containing functional groups of GO. At the same time, a broad peak appears at $2\theta = 23.06°$, representing the (002) crystal plane of carbon, indicating that there is still a part of carbon on GO that has not been oxidized [22,23].

There is no obvious strong peak in the XRD pattern of MGO, but it can still be seen that there is a broad diffraction peak corresponding to the (002) crystal plane of carbon. Compared with those of MGO and Mu, the characteristic peaks of MGMu are similar to some extent. The XRD pattern of MGMu contains both the characteristic peaks of Mu and the broad peaks of MGO. However, the peak position is slightly shifted, indicating that the layer spacing has changed. Therefore, in order to explore the change, the layer spacing is calculated according to Bragg's law (Equation (3)):

$$2d \sin \theta = n\lambda \tag{3}$$

In the XRD pattern of Mu, the peak representing the Mu (003) crystal plane is located at $2\theta = 8.900°$, from which the layer spacing can be calculated to be 0.993 nm. In MGMu, the peak of the Mu (003) crystal plane is located at $2\theta = 8.808°$, and the layer spacing is calculated to be 1.000 nm. It shows that the layer spacing had increased after the compound. With the increase of layer spacing, the force between the unit layers also decreases, so the relative sliding between the layers is more likely to occur, which is conducive to improving the lubrication effect [24].

The TEM images of GO are shown in Figure 3a,b. It can be seen that GO has a folded sheet structure. Figure 3c,d show the TEM images of freeze-dried MGO. It can be seen that, though the graphene sheets are stacked after freeze-drying, the sheet-like structure is still obvious. Figure 3e,f preset the TEM images of MGMu. The small sheet of Mu is loaded on the large sheet of MGO, which proves the successful compositing of Mu and MGO.

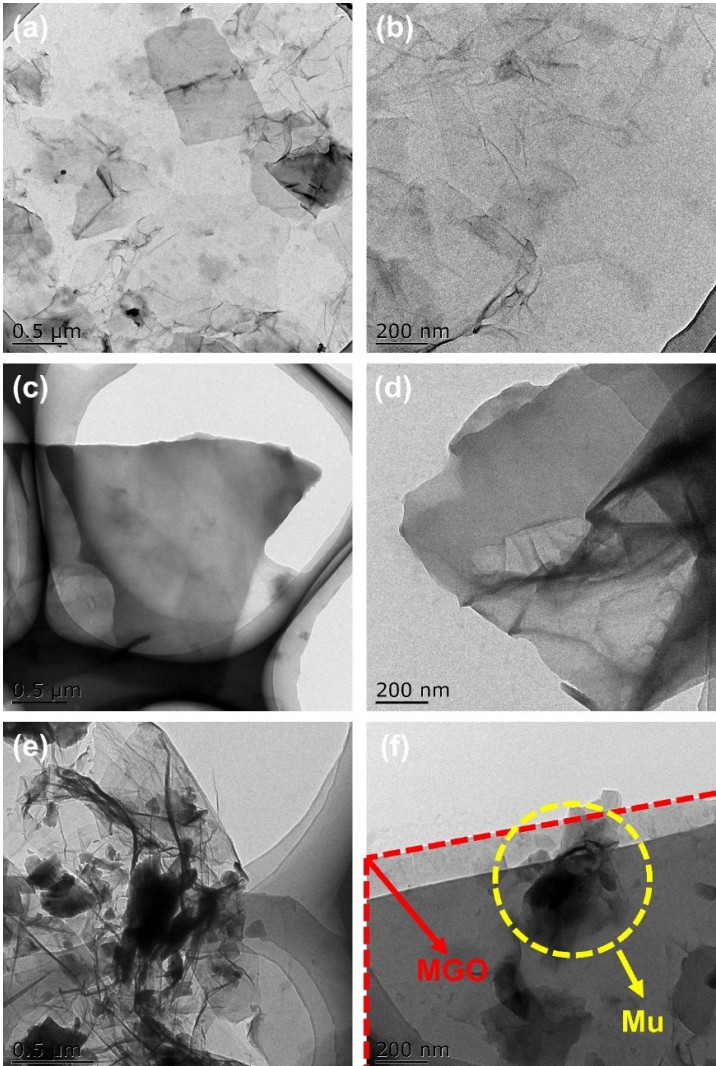

**Figure 3.** TEM images of (**a**,**b**) GO; (**c**,**d**) MGO and (**e**,**f**) MGMu.

*3.2. Friction and Wear Performance*

3.2.1. Dispersion Stability Tests

In order to evaluate the dispersion stability in oil of various samples, we conducted sedimentation tests on bare oil, GO, Mu and MGMu, to consider their dispersion stabilities in oil. The results compared to those after standing for 30 days are shown in Figure 4.

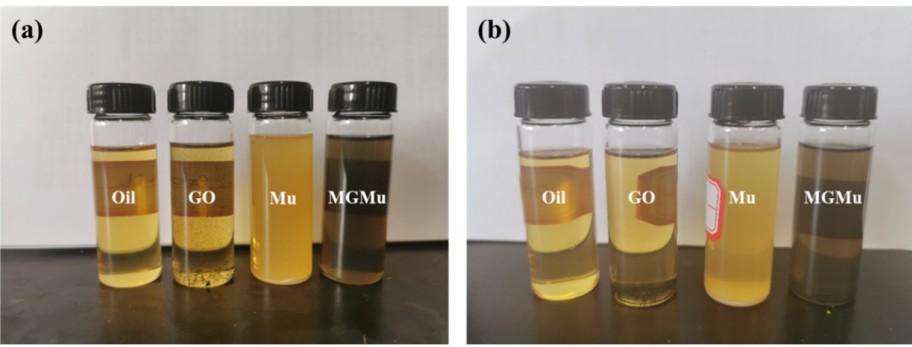

**Figure 4.** (**a**) Oil, GO, Mu and MGMu oil samples and (**b**) their pictures after standing for 30 days.

It can be seen in Figure 4a that GO is difficult to disperse in oil due to its extremely hydrophilic nature, and the particles tend to agglomerate. The Mu powder pretreated is composed of some relatively smaller particles that can be suspended in the oil, but most of the written particles still precipitate at the bottom of the bottle. However, MGMu has excellent dispersibility in oil, and there is no phenomenon of agglomeration and sedimentation at the bottom of the bottle. After standing for 30 days, as shown in Figure 4b, GO is almost completely settled, and the upper oil's color is not much different from the base oil. The oil sample of Mu still has some white turbidity, with a certain degree of transparency. At the same time, the sediment at the bottom increases, indicating that compared with the fresh sample, the Mu powder in the oil has settled a lot, and only some particles of small size can still be dispersed in it. In contrast, after 30 days, the oil sample of MGMu has no obvious sediment, indicating that its dispersion stability is good, which solves the problem of aggravating workpiece wear due to particle agglomeration to a certain extent, and provides a possibility for actual production.

### 3.2.2. Tribological Tests

The conditions of the tribological tests were described in the Experimental section. Figure 5a shows the curves of the friction coefficient of lubricating oil samples with different material concentrations of 0.4 mg/mL as a function of time. The base oil, MGO, Mu, and MGMu oil samples are compared.

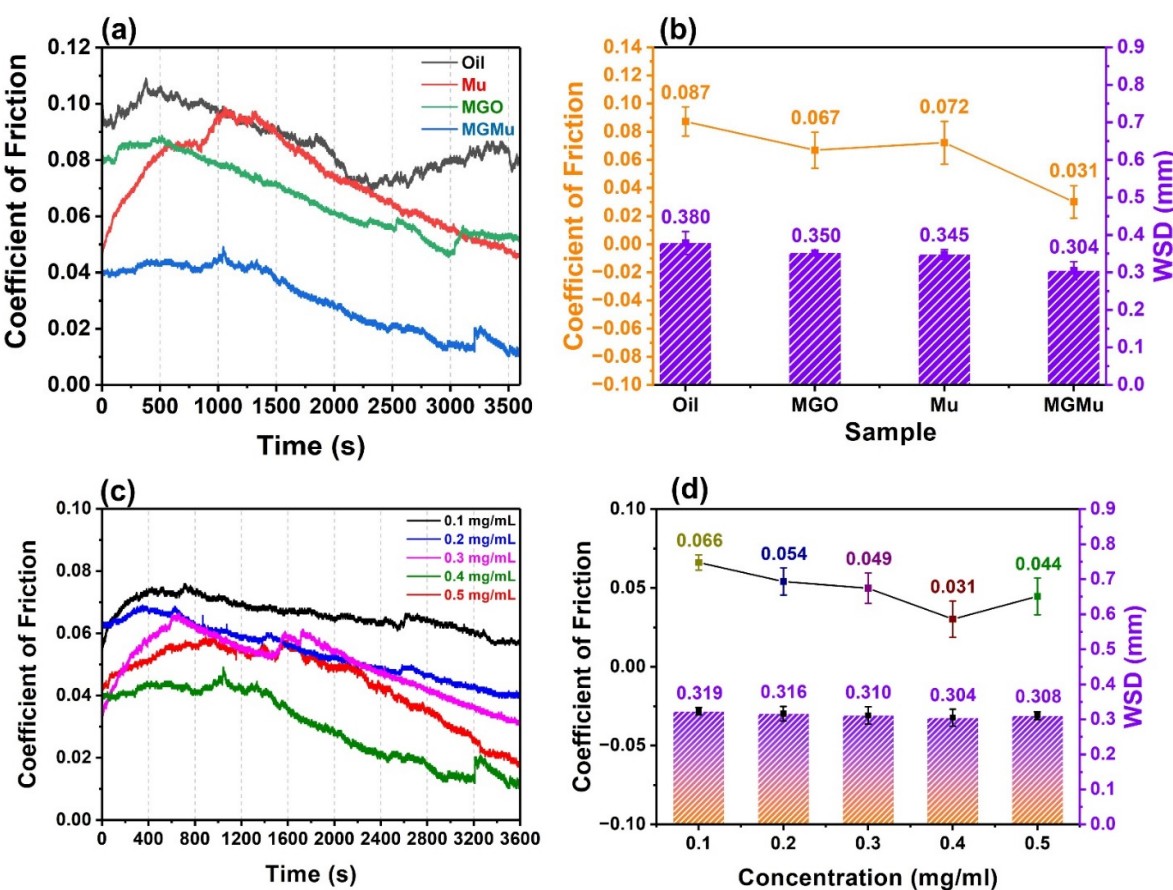

**Figure 5.** Tribological results of base oil, MGO, Mu, MGMu oil samples: (**a**) Relationship between friction coefficient and time; (**b**) Average friction coefficient and average wear scar diameter (WSD). Tribological results of MGMu oil samples with different concentrations: (**c**) Relationship between friction coefficient and time; (**d**) Average friction coefficient and average wear scar diameter.

As shown in Figure 5a, the variation characteristics of the friction coefficient curves of the four samples basically show an initial upward trend and then a downward trend.

This is because, in the initial friction stage, most asperities on the surface of the friction pair are in the state of incommensurate contact [25–27]. Adhesive contact occurs between the asperities due to the relative sliding between the friction pairs. The asperities with low mechanical strength will be destroyed and smoothed in the process, making the surface of the friction pair smoother. The destruction of the asperities requires a certain loss of mechanical energy, which makes the friction coefficient larger in the initial stage. At the same time, the oil film and the lubricating film produced by the lubricating material in the friction process have not yet formed, and the lubrication mode is mainly boundary lubrication [28–30]. As the asperities between the friction pairs gradually become smooth, and a stable oil film and lubricating film are gradually formed between the contact surfaces of the friction pairs, the ratio of the film thickness to the surface roughness gradually increases, and the lubrication mode transforms gradually from boundary lubrication to mixed lubrication. The energy consumption caused by the adhesive wear of asperities gradually decreases. Hence, the friction coefficient shows a gradually decreasing trend then, but the changes in the friction coefficients of the four samples are still quite different.

It can be seen from the figure that the friction coefficient of the base oil is large, up to 0.11, and the curve fluctuates greatly. It shows that in the friction process, the oil film formed between the friction pairs is not stable enough, causing the friction coefficient to change to show a "sawtooth" shape. The friction coefficients of MGO, Mu, and MGMu have relatively small fluctuations. The curves are relatively smooth, indicating that a relatively stable lubricating film is formed between the friction pairs during the process. After adding MGO, it can be seen that the initial friction coefficient between the friction pairs is 0.080, which is 16.7% lower than that of the base oil of 0.096. Over time, the friction coefficient gradually decreases, and the final friction coefficient is 0.052, which is 35% lower than that of base oil, indicating that MGO can improve the anti-friction performance of base oil to a certain extent.

The friction coefficient curve of Mu also shows a trend of rising first and then falling. Still, the increase of the friction coefficient in the early stage is significantly higher than that of other samples and even exceeds the friction coefficient of the base oil in the same period between 1065 and 1450 s, with the highest peak value increases by 0.097. The main reason may be that in the preparation of Mu oil samples, the Mu sample contains some relatively large particle which was not well ball-milled. In forming an oil film in the friction pair, the excessively large particles were not only unfavorable for filling into the oil film between the friction pairs but also hindered the friction pair's relative sliding, so that it made the friction coefficient increase [13]. However, during the friction process, part of Mu with a larger particle size settled at the bottom of the oil tank under the action of gravity. The other part was gradually broken into small particles which were easily filled into the oil films between the friction pairs under the mechanical force of the rotation and extrusion of the friction pairs, playing a lubricating role. It can also be proved by the fact that the friction coefficient curve begins to drop sharply in the second half of the curve. Finally, the friction coefficient decreases to 0.046, which is 42.5% lower than that of the base oil, proving that Mu has a certain anti-friction performance.

It can be seen from Figure 5a that the lubricating performance of the MGMu oil sample is the best, for the initial friction coefficient is only 0.039. The final friction coefficient is only 0.012, which is 59.3 and 85.0% lower than that of the base oil, respectively, which proves that the prepared MGMu has excellent friction reduction and anti-friction performance.

Wear scar diameter (WSD) is also an indicator for evaluating the lubricating performance of lubricating materials [31]. The average value and standard deviation of friction coefficients of different samples and the average wear scar diameter (WSD) of friction pair steel balls after testing is demonstrated in Figure 5b. As shown in Figure 5b, the average friction coefficients of MGO, Mu and MGMu oil samples were 0.067, 0.072 and 0.031, respectively, and compared with that of the base oil, the degrees of decline were 23.0, 17.2 and 64.4%, respectively.

The average WSD on the surface of the steel balls rubbed by the base oil is 0.380 mm, and after adding additives of different compositions to the oil, the wear scar diameters of the steel balls all decreased to some degrees. After adding MGO and Mu, compared with the base oil, the average WSD is reduced by 7.8% (0.350 mm) and 9.2% (0.345 mm), respectively, indicating that both MGO and Mu can improve the wear resistance of the friction pair. The average WSD of the MGMu oil sample steel ball is the smallest, only 0.304 mm, reduced by 10.5%. This result indicates that the wear resistance of MGMu is better than that of the single components. The main reason may be the introduction of the additives and the synergistic effect between the two components. Compared with the base oil, the above three oil samples containing the additive component all reduce the wear of the friction pair to a certain degree. The reason can be explained that the additive components react with the elements on the surface of the friction pair during the friction process to form different repair layers and fill the cracks and furrows of the steel ball due to the wear. This reduces the degree of wear on the surface of the steel ball macroscopically, making the wear scar relatively small. It is worth noting that in the single-component oil samples. However, the average friction coefficient of the Mu oil sample is higher than that of the MGO oil sample; the average wear scar diameter is slightly smaller than that of the MGO oil sample. It confirms that the repairing layer formed during the friction process of Mu oil has better wear resistance. Therefore, the good wear resistance of MGMu may be mainly attributed to Mu. Therefore, further analysis of the wear scar was carried out to explore the formation mechanism and the components of the repairing layer of MGMu.

The above analysis shows that MGMu has excellent lubricating properties. Figure 5c also demonstrates the tribological test results of MGMu oil samples with different concentrations (0.1, 0.2, 0.3, 0.4, and 0.5 mg/mL). Figure 5a shows the curves of friction coefficients as a function of time for samples with different concentrations. It can be seen that no matter how much of MGMu is added, the friction coefficients are lower than that of the base oil, and the coefficients of friction all decrease with time. When the concentration of MGMu is 0.3 mg/mL, the curve fluctuates, and the friction coefficient rises sharply at the initial stage, indicating that the oil sample may contain insufficient ball-milled Mu. The large particle size will increase the surface roughness of the friction pair when the oil film forms. This causes Mu to increase the resistance during the relative movement of the friction pair in the initial crushing process. As the concentration is gradually increased to 0.4 mg/mL, the final friction coefficient gradually decreases. It indicates that as the content of MGMu in the oil film gradually increases, MGMu is more filled between friction pairs, reducing the number of direct contact asperities between the friction pairs. That is to reduce the microscopic actual contact area (surface roughness) between the friction pairs, thereby reducing the friction coefficient. However, when the concentration of MGMu is increased to 0.5 mg/mL, compared with that of 0.4 mg/mL, the friction coefficient becomes larger as a whole. This is because when the content of MGMu in the oil film is too large, the adsorption and agglomeration of MGMu between the asperities may occur, even if the probability of contact between the friction pair itself is reduced. The MGMu itself will agglomerate into large particles and form a kind of "asperity," increasing the surface roughness of the friction pair, thereby increasing the friction coefficient.

Figure 5d shows the average friction coefficients and average WSDs of oil samples with different concentrations. It can be seen that the variation of the average friction coefficient with the concentration is consistent with the above analysis. When the concentration of MGMu added is 0.1, 0.2, 0.3, 0.4 and 0.5 mg/mL, the average friction coefficient is 0.066, 0.054, 0.049, 0.031 and 0.044, respectively, compared with that of the base oil of 0.087, decreases by 24.1, 37.9, 43.6, 64.4 and 49.4%. At the same time, the average WSD is found to be 0.319, 0.316, 0.310, 0.304 and 0.308 mm, which is 16.1, 16.8, 18.4, 20.0 and 18.9% smaller than that of the base oil (0.380 mm), respectively. It can be seen that with the change of concentration, although the changing trend of the average wear scar diameter is the same as that of the friction coefficient, the difference in the value of the change is not large. This means that no matter how much the added amount is in the range of considered

concentration, as long as there is MGMu in the oil film during the friction process, the degree of wear on the surface of the friction pair can be reduced.

In order to explore the anti-wear effect of MGMu, a series of characterizations on wear scars were carried out subsequently.

### 3.2.3. Wear Scar Analysis

Figure 6 shows the SEM and EDS images of wear scars with different additives. The SEM image of the wear scar in the base oil is shown in Figure 6a. Although the scratches on the surface of the wear scar are not deep, there are still many defects and a structure similar to "pits" on the surface of the wear scar. It shows that, in the process of friction, although there is an oil film composed of base oil, the lubrication mode is mainly boundary lubrication. That is, the friction pair has more contact with the asperities, and the boundary lubricating film formed by the interaction between the base oil and the surface of the friction pair bears almost all the load [32–34]. During the rubbing process, the asperities stick to each other and are damaged, resulting in defects similar to "pits" on the surface of the wear scar.

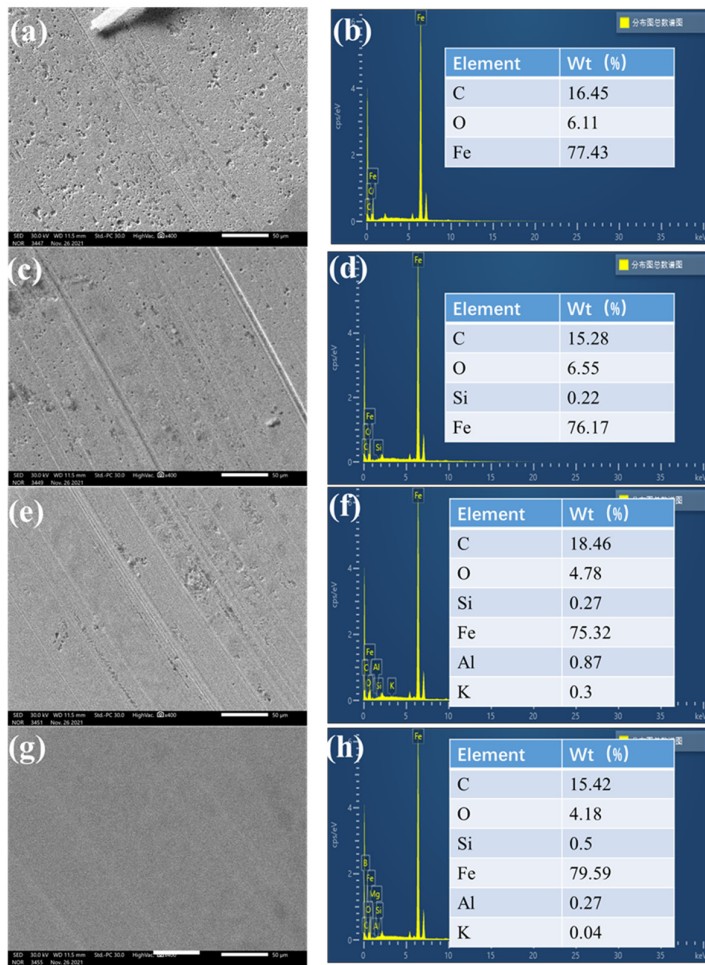

**Figure 6.** SEM and EDS images of wear spots of (**a**,**b**) oil; (**c**,**d**) MGO; (**e**,**f**) Mu; (**g**,**h**) MGMu.

However, for the oil sample containing MGO, Mu or MGMu, their wear scars are shown in Figure 6c,e,g. The surface of the wear scar is smoother than that in the base oil except for some scratches. It is worthy to note that for the MGMu oil sample, its wear scar exhibits very smooth surface, even no obvious scratches or obvious "pits" observed.

Due to the presence of additive components in the oil film between the friction pairs during the friction process, the lubrication mode changes from boundary lubrication mode to the mixed lubrication mode. The additive blocks the surface of the friction pair to a

certain extent. The load is borne by both the boundary lubricating layer and the additive components, instead of only the boundary lubricating layer formed by the friction pair and the base oil, thereby reducing the wear between the friction pairs and reducing the "pit" defects on the wear scar surface. Additionally, comparing the wear scars of different additives, it can be seen that the "pits" in the wear scar for the Mu sample are less than that of MGO. It indicates that Mu is more effective in reducing the wear degree than MGO, consistent with the previous analysis results for the average wear scar diameter of different samples. This result verifies that Mu forms a repairing layer on the surface of the wear scar that can better fill the surface defects of the wear scar during the friction process. From the wear scars of the MGMu oil sample, it can be seen that the surface of the wear scars has no obvious scratches and no obvious "pits". It proves that MGMu has excellent anti-wear property and forms a better self-repairing layer than the single-component additive Mu and MGO.

EDS confirmed the chemical contents of the wear scar surface. It can be seen from Figure 6b,d,f,h that, after adding different additives, the elements on the surface of the wear scars are significantly different.

The surface of the base oil wear scar mainly contains C, Fe, O elements. For the MGO sample, the surface of its wear scar contains a small amount of Si element besides C, Fe, O. It indicates that a small amount of MGO is attached to the surface of the wear scar and fills the surface of the wear scar to form a self-repairing layer. As for the Mu sample, there are not only C, Fe, O, but also K, Si, Al and other elements detected on the surface of Mu wear scar. The existence of K, Si, and Al is basically attributed to Mu, indicating that a self-repairing layer is formed on the surface of the wear scar during the tribological test. The generation of this repairing layer may be due to some tribochemical reaction of Mu during the friction process [1]. As for the MGMu sample, the EDS result of its wear scar also presents C, O, Fe, Si, Al and K elements, which can confirm the existence of a self-repairing layer generated from MGMu.

Together with the SEM image of MGMu, one can infer the self-repairing layer from MGMu must be different from others. Therefore, the hardness tests and XPS measurements were used to analyze the wear scars further.

The above SEM and EDS results show that the additive components can fill the defects of the wear scar surface to some certain extent. And the defects are attributed to uneven scratches on the surface caused by the plastic deformation of asperities adhering to each other during the relative motion. In the theory of adhesion, Heilmann and Rigney proposed the energy model of friction. In this model, the friction coefficient is expressed in Equation (4) as follows [35]:

$$f_g = \frac{A_r}{W} \tau_{max} f\left(\frac{\tau_s}{\tau_{max}}\right) \tag{4}$$

In Equation (4), $A_r$ represents the real contact area of the two surfaces in contact with each other, $f_g$ represents the furrow friction coefficient, $W$ represents the load, $\tau_{max}$ and $\tau_s$ represent the ultimate shear force of the friction pair material, and the average shear force of interaction surface, respectively. The calculation formula (Equation (5)) of $f\left(\frac{\tau_s}{\tau_{max}}\right)$ in Equation (4) is as follows:

$$f\left(\frac{\tau_s}{\tau_{max}}\right) = 1 - 2\frac{\ln\left(1 + \frac{\tau_s}{\tau_{max}}\right) - \frac{\tau_s}{\tau_{max}}}{\ln\left[1 - \left(\frac{\tau_s}{\tau_{max}}\right)^2\right]} \tag{5}$$

Since the upper and lower steel balls are of the same material in this experiment, the shear force of the contact surface can be approximately regarded as the shear force of the actual contact asperities, that is, the shear force of the material itself. Therefore, the following reasonable assumption can be made as follows in Equation (6):

$$\tau_s \approx \tau_{max} \tag{6}$$

Therefore, $f\left(\frac{\tau_s}{\tau_{max}}\right) \approx 1$, and $A_r$ is a value that is difficult to measure in practical applications. But due to the friction mode of asperities acts similarly to the indenter in a microhardness tester, and Hardness $H$ is defined as the ratio of load to the base area, that is: $\frac{W}{Ar}$, so $\frac{Ar}{W}$ in the previous formula can be replaced by $\frac{1}{H}$, resulting in the following formula (Equation (7)):

$$f_g = \frac{1}{H}\tau_{max} \qquad (7)$$

It can be seen that the wear scar hardness is inversely proportional to the friction coefficient; that is, the greater the hardness, the smaller the friction force. Although in the friction process, the force on the asperities not only comes from the load's normal pressure and the shear force generated by the relative movement of the adhered asperities, the effect is not exactly like that of the indenter in the hardness tester. However, a rational assumption can be made here, approximately equaling the actual contact area between the asperities to that of the indenter, to convert the actual contact area that is difficult to measure into a microhardness value that can be directly measured. Equation (7) can be used here to evaluate the self-repairing film on the wear scars in our work.

The micro indentation hardness values of the wear scars of different oil samples after friction tests are shown in Figure 7. The result in Figure 7 shows that the microhardness of the wear scars is improved when additives of different components are added to the base oil. Compared with the base oil, the microhardness of the wear scar in MGO, Mu and MGMu oil samples was increased by 12.9, 13.4 and 16.1%, respectively. The results of hardness tests show that the oil samples of MGMu have the largest wear scar hardness of the steel balls. It indicates further that MGO and Mu may have a synergistic effect in this oil sample, forming a self-repairing layer with a higher hardness different from that of the single component.

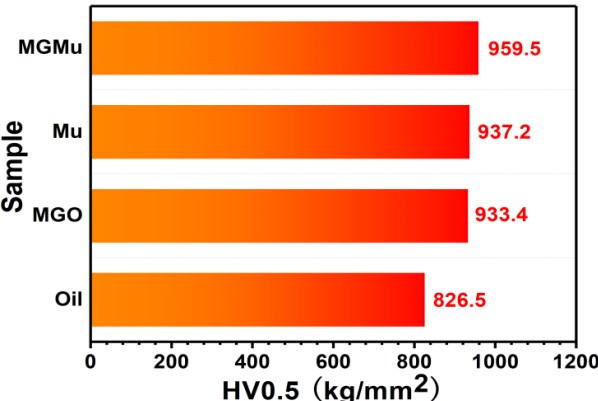

**Figure 7.** Microhardness of wear scars of base oil, MGO, Mu and MGMu oil samples.

In order to further explore the composition and formation mechanism of the self-repairing layer, XPS measurements were carried out for the wear spot surface of MGMu, and the results are shown in Figure 8. The XPS full spectrum of the wear scar contains elements such as C, O, Si, K, Al and Fe (Figure 8a), further confirming the presence of components of MGMu in the self-repairing film. It is consistent with the result of EDS analysis.

The C1s energy spectrum in Figure 8b presents the deconvoluted peak at 285.2 eV, which is attributed to the C–C in the internal structure of MGMu [35–37], proving that a small amount of modified graphene oxide may be attached to the surface of the wear scar. At the same time, the peak at 101.2 eV in the Si2p spectrum (Figure 8c) assigned to organic Si also proves the presence of silane coupling agent in MGMu. The peak at 102.3 eV (Figure 8c) is attributed to aluminosilicate of Mu. All the analyses prove the presence of MGMu on the surface of the wear scar.

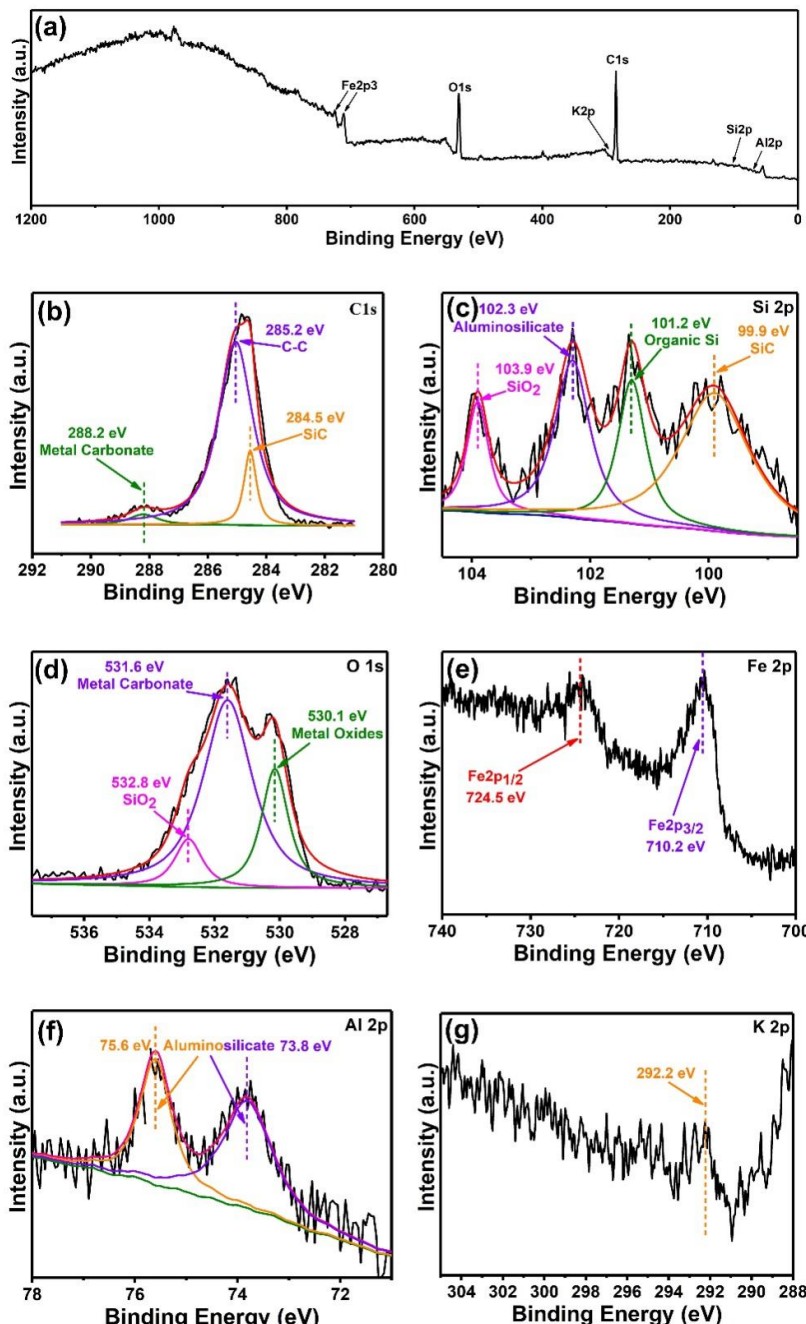

**Figure 8.** XPS results of MGMu wear scar. (**a**) full spectrum; (**b**) C1s; (**c**) Si2p; (**d**) O1s; (**e**) Fe2p; (**f**) Al2p; (**g**) K2p.

Moreover, the peak of C1s at 284.5 eV (Figure 8b) corresponds to the binding energy of C–Si in SiC, and the peak at 99.9 eV in the spectrum of Si2p (Figure 8c) is also attributed to the Si–C in SiC. At the same time, the peak at 103.9 eV reflects the existence of Si–O in $SiO_2$, and the peak at 532.8 eV in the O1s energy spectrum (Figure 8d) can also prove the existence of $SiO_2$ on the surface of the friction pair, which reflects the binding energy of O–Si in $SiO_2$. Therefore, the results confirm the formation of SiC and $SiO_2$ during the process of friction in the presence of MGMu as an oil additive. These chemical compounds may be produced in some physical and chemical reactions that occurred in MGMu under friction [5]. SiC is a non-metallic ceramic material with high hardness, and $SiO_2$ also has the effect of increasing hardness [38]. The hardness test results show that the wear scar hardness is the highest after adding MGMu to the oil, which may be due to the generation of SiC and $SiO_2$.

The peak of O1s at 531.6 eV corresponds to metal carbonates [39], and the peak at 288.2 eV of C1s also indicates the existence of O–C=O in metal carbonates [40], which may be generated in the friction process. It can be seen from the Fe2p spectrum (Figure 8e) that there are two peaks located at 724.5 and 710.2 eV, respectively, which proves the existence of $Fe_2O_3$ or $Fe_3O_4$ on the surface of the wear scar. It can be inferred that the peak located at 530.1 eV in the O1s spectrum reflecting metal oxides corresponds to –Fe(III)–O–. It shows that during the friction process, the Fe on the surface of the steel ball of the friction pair underwent an oxidation reaction [41,42].

As mentioned above, muscovite is a TOT-type layered silicate. The bonding between the unit layers is ionic bonding. Compared with the bonding force between the silicon-oxygen tetrahedron and the aluminum oxygen octahedron in the layer, it is weakened, and the sliding between layers easily occurs, such that Mu has a certain lubricating capacity [43]. Figure 8g shows that the peak at 292.2 eV shows the existence of the K element, which proves that during the friction process, the Mu of MGMu experienced interlayer-sliding and released the $K^+$ between the layers.

Meanwhile, the peaks at 75.6 and 73.8 eV in the Al2p spectrum (Figure 8f) and the peak at 102.3 eV in Si2p indicate the presence of aluminosilicates on the surface of the friction pair. Since there are two peaks in Al2p, it shows that there is more than one type of aluminosilicate on the surface. There may be a small amount of Mu in MGMu, and new aluminosilicates may be generated due to the easy exchange of metal ions in the aluminum oxide tetrahedron between the layers of Mu itself.

In addition, as mentioned above, the asperities will release energy during the process of destroying and smoothing, and Mu may also react to generate new silicates during the friction process. The above-mentioned friction reaction mechanism is analyzed in detail subsequently.

### 3.3. Lubricating Mechanism Analysis

The previous analysis shows that MGMu can form a self-repairing layer with a high hardness on the friction surface during the friction process [1,44–46]. In order to explore the reaction process, the lubrication mechanism is analyzed and explained here, combined with the mineralogy principle and the adhesion theory of friction.

Figure 9 is the schematic diagram of the lubrication mechanism of the composite materials MGMu. Figure 9a,b present the microscopic schematic diagrams of the friction pair surface. During the friction process, the composite nanosheets in the lubricating oil are gradually and evenly distributed on the surface of the friction pair under the action of mechanical force. The process forms a stable lubricating oil film, thereby avoiding the direct contact of some friction pair asperities to some extent and reducing the wear of the workpieces. Both layered silicate and graphene oxide are two-dimensional layered materials. The interlayer bonding force is weak, so the interlayers are easy to slide to play a lubricating role, which can effectively reduce the friction coefficient, as shown in Figure 9d. The EDS analysis of the MGMu wear scar showed the existence of the K element, which proved that the interlayer $K^+$ was released due to the interlayer sliding.

According to the oxidation friction theory, during the friction process, under the mechanical action of the relative motion, the friction pair produces many lattice defects on the metal surface, and the existence of lattice defects provides active sites for the friction reaction of the friction pair. That is, the electrons on the metal surface overflow through thermal excitation and tunnel effect, the oxygen adsorbed on the surface of the friction pair absorbs them to form oxygen ions ($O^{2-}$), and the metal on the surface is oxidized to form metal cations. The oxygen ions and the metal cations diffuse into each other and react to form metal oxides when they meet [47,48], as shown in Figure 9c. This shows that the metal elements contained in the friction pair are ionized to form $Fe^{3+}$ on the surface and react with the $O^{2-}$ adsorbed on the surface of the friction pair to produce $Fe_2O_3$. The structure of Mu is shown in Figure 9e, and the Al atoms of the aluminum-oxygen octahedron in the Mu layer can be replaced by other metal atoms [49]. When the asperities on the surface of the

friction pair are plastically deformed and damaged, they will release energy and generate flash temperature so that the Al atoms in Mu will exchange with the Fe atoms in the metal to generate new aluminosilicates [5,50]. They release $SiO_2$, which will further react with the carbon of graphene oxide in the composite to form SiC [7], thereby increasing the surface hardness of the wear scar. The reaction formula is shown in Equation (8):

$$SiO_2 + 3C = SiC + 2CO \tag{8}$$

The resulting new aluminosilicate will fill in the defects on the wear surface, thereby endowing MGMu with self-repairing ability.

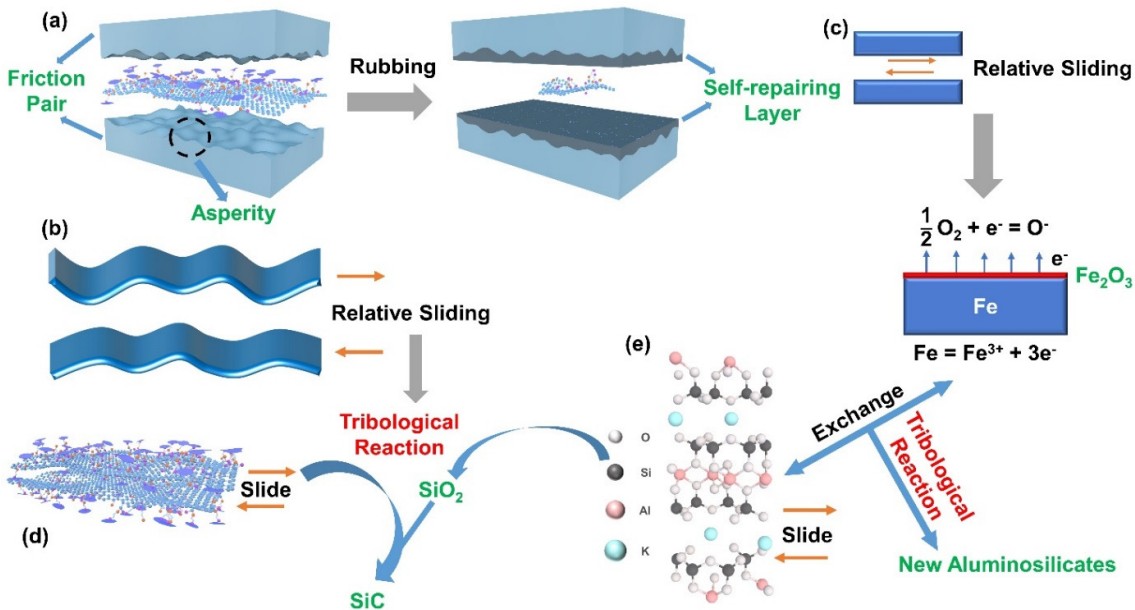

**Figure 9.** Schematic diagram of lubrication mechanism: (**a**) lubrication model of friction pair; (**b**) asperity model; (**c**) schematic diagram of oxide film formation mechanism; (**d**) schematic diagram of composite structure; (**e**) schematic diagram of muscovite structure.

In the adhesion theory, if a tangential force $F$ is applied to the upper surface of the contact material, the actual contact area will grow laterally; that is, the adhesive point will have a lateral expansion. And when the resultant force of tangential stress and normal stress reaches the material yield limit, plastic flow occurs at the adhesive point. Suppose (Equation (9)):

$$\sigma^2 + a\tau^2 = K^2 \tag{9}$$

$a$ and $K$ are undetermined values. Among them, $\sigma$ is the normal stress, perpendicular to the section direction; $\tau$ is the tangential stress, parallel to the section direction; $K$ is the resultant stress of the two stresses. Due to $\sigma = \frac{W}{A_r}$, $\tau = \frac{F}{A_r}$, Equation (9) can be written as:

$$\left(\frac{W}{A_r}\right)^2 + a\left(\frac{F}{A_r}\right)^2 = K^2 \tag{10}$$

$W$ means the normal force, $F$ means the tangential force, and the contact area under the combined action of $W$ and $F$ is represented by $A_r$. When the resultant stress reaches the compressive yield limit of the material (Equation (11)):

$$K^2 = \sigma_y{}^2 = \left(\frac{W}{A_r}\right)^2 + a\left(\frac{F}{A_r}\right)^2 \tag{11}$$

$\sigma_y$ is the plastic flow pressure (yield pressure) of the metal, approximately equal to the hardness value $H$. At this time, plastic deformation occurs on the real contact area, and the area does not continue to expand.

When discussing the adhesion theory of friction, it is generally assumed that the friction surface is clean. But in a normal atmosphere, metal surfaces are always covered by oxide films or other contamination films. Therefore, the friction of such a metal friction pair is actually the friction of the oxide film. The metal-to-metal friction can be directly formed only after the oxide film is destroyed. Moreover, the presence of oxide films or other adsorption films and chemical reaction films on the metal surface is advantageous from the viewpoint of friction and wear. Because of the interface layer formed by the oxide film or other contamination film between the two surfaces, the critical shear stress $\tau_f$ of the film is smaller than the critical shear stress $\tau_c$ of the metal adhesion point, that is (Equation (12)):

$$\tau_f = C\tau_c, \ 0 < C < 1 \tag{12}$$

When the tangential stress $\tau < \tau_f$ the normal stress and tangential stress can be transmitted to the metal matrix through this interface layer so that the plastic flow occurs; that is, the adhesion point increases, and the contact area increases.

When $\tau = \tau_f$, the interface layer is sheared and starts to slide. When $F$ is large, that is, $\frac{F}{A_r} \gg \frac{W}{A_r}$, then according to Equation (11), $\frac{W}{A_r}$ can be ignored, so Equation (13) is presented:

$$\sigma_y{}^2 \approx a\left(\frac{F}{A_r}\right)^2 = a\tau_c{}^2 \tag{13}$$

When the resultant stress reaches the compressive yield limit of the material, plastic deformation occurs at the contact points of the surface film. So, Equation (9) can be drawn as:

$$K^2 = \sigma^2 + a\tau_f{}^2 = \sigma_y{}^2 \tag{14}$$

Substitute Equations (12) and (13) into Equation (14), simplified to get:

$$\frac{\tau_f}{\sigma} = \frac{C}{a[(1-C^2)]^{\frac{1}{2}}} \tag{15}$$

According to the law of friction, the coefficient of friction $f$ can be expressed as:

$$f = \frac{F}{W} = \frac{\tau_f A_r}{\sigma A_r} = \frac{C}{a[(1-C^2)]^{\frac{1}{2}}} \tag{16}$$

It can be known from Equation (16) that when $C \rightarrow 1, f \rightarrow \infty$, this is inconsistent with the actual situation. But obviously, if the critical shear strength $\tau_f$ of the film is equal to the shear strength $\tau_c$ of the base metal, the coefficient of friction is at most equal to the coefficient of friction of the base metal and cannot become infinite. It can be seen that the theory is still imperfect, but it can still explain some problems.

The relationship curve of $f \sim C$ can be obtained from different values of $a$, as shown in Figure 10. It can be seen that as $C$ decreases from 1, the $f$ value decreases rapidly, which indicates that as long as there is a thin film at the interface, the connection strength of the interface will be greatly weakened, resulting in a sharp decrease in the value of the friction coefficient $f$.

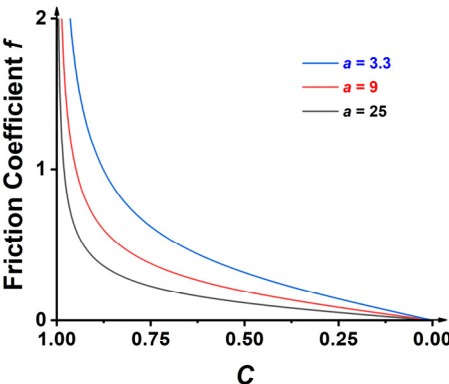

**Figure 10.** Relationship curve of $f \sim C$ with different values of $a$.

When $C$ is small, Equation (16) can be written as $f = \frac{C}{\sqrt{a}}$, and according to Equation (13), $\sqrt{a} = \frac{\sigma_y}{\tau_c}$. So:

$$f = \frac{C\tau_c}{\sigma_y} = \frac{\tau_f}{\sigma_y} = \frac{\text{Shear strength limit of interfacial films}}{\text{Yield pressure of base metal}} \tag{17}$$

Due to the friction interface's small shear strength, the adhesion point area has not had time to increase significantly during sliding, and the interface film is sheared. In addition, the lateral growth of the adhesion area during sliding mainly depends on the plastic deformation of the base metal. But due to the inhibition of the interface film during the deformation, the lateral growth of the contact area is much smaller than that in the direct contact of metal-to-metal; that is, the frictional force is much smaller. At the same time, as mentioned above, in the friction process, the friction pair will produce the $Fe_2O_3$ oxide layer. The Al atoms in MGMu will exchange with the Fe atoms in the metal to generate new aluminosilicates and $SiO_2$, a chemically reactive film with lower shear strength, which makes MGMu's oil samples have a lower coefficient of friction.

## 4. Conclusions

This work successfully prepared the lubricant additive MGMu with Mu and the graphene oxide modified by a silane coupling agent. The prepared MGMu has good lipophilicity and can be stably dispersed in a base oil for 30 days with almost no sedimentation. MGMu exhibits an excellent lubricating property compared with MGO and Mu. MGMu, MGO and Mu as additives can improve the lubricating performance of the base oil, and compared with the base oil, the average friction coefficient decreases by 64.4, 23.0 and 17.2%, respectively. The average WSD for MGMu, MGO and Mu oil samples is also reduced by 20.0, 7.8 and 9.2% compared with the base oil. When the addition amount of MGMu is at a very low concentration (the mass fraction is about 0.01~0.06%, that is, 0.1~0.5 mg/mL), the lubricating performance of the lubricating oil can be improved. The lubricating performance is the best when the concentration of MGMu is 0.4 mg/mL. Different additives contribute to the formation of the self-repairing layer, improving the hardness on the surface of the wear scars, thereby filling the "pit" defects caused by the wear of the asperities. The self-repairing effect of MGMu exhibits the best with the highest hardness of wear scar. The wear scar surface of the corresponding sample basically has no obvious scratches and other structural defects.

According to the analysis of the chemical composition of MGMu wear scar, the composition of the self-repairing layer was explored, and a possible formation mechanism was proposed. During the friction process, some chemical reactions between MGMu and some Fe on the surface of the friction pair result in the generation of new aluminosilicates, $SiO_2$, SiC and iron oxides, which have significant effects on improving the surface hardness of wear scars. Besides, according to the adhesion theory of friction, the chemically reactive film above has lower shear strength, which makes MGMu's oil samples have a lower coeffi-

cient of friction. This work provides a new path for developing novel lubricant additives and the view to understand the mechanism of self-repairing in the field of lubrication.

**Supplementary Materials:** The following supporting information can be downloaded at: https://www.mdpi.com/article/10.3390/lubricants10080190/s1, Figure S1: The unit layer structure of Muscovite; Figure S2: The friction pair model of the four-ball friction tester.

**Author Contributions:** Conceptualization, Q.H.; Funding acquisition, Q.H.; Investigation, Z.Z., Y.L. and Q.H.; Methodology, W.L. and Q.H.; Project administration, Q.H.; Resources, W.L.; Writing—original draft, Z.Z.; Writing—review & editing, Y.L., W.L. and Q.H. All authors have read and agreed to the published version of the manuscript.

**Funding:** This research was funded by the "Fundamental Research Funds for the Central Universities of China," grant number No.30922010502. We also thank the support of the Analysis and Test Center, Nanjing University of Science and Technology, for XRD data collection.

**Data Availability Statement:** Not applicable.

**Conflicts of Interest:** The authors declare no conflict of interest.

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
