# Peer review of "Modified Graphene/Muscovite Nanocomposite as a Lubricant Additive: Tribological Performance and Mechanism"

_lubricants, doi:10.3390/lubricants10080190_

Round 1
Reviewer 1 Report
Tribological performance of modified Graphene/Muscovite Nanocomposite.
Interesting topic and well organized manuscript.
Good structured introduction section. I recommend recent references (>2020)
Figure 1 is too small...
In the tribological evaluation, what standard was followed for those testings? why the parameters detailed in the manuscript were chosen? any particular reason? are those coming from an industrial emulation / simulation?
What was the process to perform the tribological evaluations? were the samples sonicated before the testings? how was the lubricant deposited in the container/holder with the stationary steel balls?
Figure 5 is small, not clear for the reader... please fix it or arrange it to be bigger. Thanks for incorporating the error bars to COF and WSD results!
Good analysis for WSD (Figure 6).
About the lubrication mechanism. In line 536 you mentioned the following "self-repairing layer with high hardness on the friction surface during the friction process", is there any indication about this phenomena is attributed to the MGMu effect? are there any references from literature that are based on this? Great schematic in figure 9 by the way!
Reviewer 2 Report
The authors have prepared an interesting manuscript of a new composite nanomaterial with promising antiwear applications. They have gone in depth with the friction mechanism and I only have some minor comments/suggestions.
1. The structural characterization of the GO-Mu composite is complicated since they have used FTIR instead of Raman spectroscopy. Raman will give much better information about the defects of GO when the muscovite is added. Furthermore, since this is a two steps modification, they can easily follow the changes on GO.
2. Different concentrations of nanomaterial were dispersed in the lubricating oil and a series of new nanofluids were obtained. I miss the characterization of these nanofluids: rheology, dielectric constant, FTIR, among other techniques. (DOI:10.3390/nano10030535). The viscosity measurements can also help with the tribological data.
Reviewer 3 Report
In this present manuscript, the modified graphene/muscovite nanocomposite was prepared, the tribological performance and mechanism of MGMu oil sample was discussed. The work is interesting, but some minor revisions are made before formally acceptance.
1. English language and style are minor spell check required.
2. The introduction can be improved to provide sufficient background and some recently relevant references are supplied.
3. According to which test standards, the tribological properties were mainly measured under 197 N with a rotary velocity of 600 r/min at room temperature for 1h? What is the relationship between the tribological mechanism and the calculation of the contact pressure of the sphere at F=197N?
4. It would be better that the morphology of the material be observed by SEM, and the chemical composition of the materials surface be analyzed by XPS.
5. The C1s energy spectrum in Figure 8b presents the deconvoluted peak at 285.2 eV is attributed to the C-C? The peak of C1s at 284.5 eV (Figure 8b) corresponds to the binding energy of C-Si in SiC? They are 284.7eV and 285.3eV in the figure 8.
